

# Observation of the process of snow accumulation on the Antarctic Plateau by time lapse laserscanning

Ghislain Picard[1], Laurent Arnaud[1], Romain Caneill[1,*], Eric Lefebvre[1], and Maxim Lamare[1]

[1]UGA, CNRS, Institut des Géosciences de l'Environnement (IGE), UMR 5001, Grenoble, F-38041, France
[*]Now at: Department of Marine Sciences, University of Gothenburg, Gothenburg, Sweden

**Correspondence:** Ghislain Picard (ghislain.picard@univ-grenoble-alpes.fr)

**Abstract.** Snow accumulation is the main positive component of the mass balance in Antarctica. In contrast to the major efforts deployed to estimate its overall value on a continental scale – to assess the contribution of the ice-sheet to sea-level rise – knowledge about the accumulation process itself is relatively poor, although many complex phenomena occur between snowfall and the definitive settling of the snow particles on the snowpack. Here we exploit a dataset of near-daily surface

elevation maps recorded over three years at Dome C using an automatic laserscanner sampling $40$–$100\,\mathrm{m}^2$ in area. We find that the averaged accumulation is relatively regular over the three years at a rate of $+8.7\,\mathrm{cm\,yr}^{-1}$. Despite this overall regularity, the surface changes very frequently (every 3 days on average) due to snow erosion and heterogeneous snow deposition that we call accumulation by "patch". Most of these patches (60–85%) are ephemeral but can survive a few weeks before being eroded. As a result, the surface is continuously rough (6–8 cm root mean square height) featuring meter-scale dunes aligned

along the wind and larger, decameter-scale undulations. Additionally, we deduce the age of the snow present at a given time on the surface from elevation timeseries and find that snow age spans over more than a year. Some of the patches ultimately settle, leading to an heterogeneous internal structure which reflects the surface heterogeneity, with many snowfall events missing at a given point, whilst many others are over represented. These findings have important consequences for several research topics including surface mass balance, surface energy budget, photochemistry, snowpack evolution and the interpretation of

the signals archived in ice cores.

## 1 Introduction

The accumulation of snow and ice on the Antarctic ice-sheet is a major component of its mass balance. Many studies aim to estimate the accumulation rate at regional or continental scales. They use in-situ observations and interpolation (Favier et al., 2013), various satellite observations (Vaughan et al., 1999; Arthern et al., 2006), regional and global climate models (Krinner

et al., 2006; Lenaerts et al., 2012; Agosta et al., 2018), and ever more a combination of these. A relative consensus on the present time annual accumulation has been reached (the IMBIE team, 2018). However, this is not the case for future projections, which can only rely on climate modelling. Even though the models are extensively evaluated against current observations, they are biased and some processes may have a different influence in the future, potentially reducing the model's skills.



This important attention paid to the accumulation quantification contrasts with the limited number of investigations on the process of accumulation itself. This process appears to be more complex in Antarctica compared to other regions (mountains, sea-ice, ...). Snowfall is the main process of mass gain for the surface, but the direct condensation of water vapour on and in the snowpack can be significant as well (Krinner et al., 2006), although this is debated (Agosta et al., 2018). Sublimation during
snowfall and from the surface after deposition is the main process of mass loss, but it is associated with large uncertainties highlighted by the very wide range of estimates in the literature obtained by modeling, from a few percent to half of the precipitation amount (Déry and Yau, 2002; Lenaerts et al., 2010, 2012; Agosta et al., 2018). Moreover some recent in-situ and remote sensing observations (Gallet et al., 2014; Grazioli et al., 2017) have revealed underappreciated sublimation components that are not yet present in models. Transport of snow by wind has a minor direct contribution on the overall SMB of the
ice-sheet because the advection of snow out of the ice-sheet to the ocean represents a negligible fraction. It however has an indirect effect by promoting airborne snow sublimation which is thought to be significant (Gallée et al., 2001) but still uncertain (Sharma et al., 2018). Transport also has an increasingly important role on the snow distribution as the scale of interest get smaller.

At the meter scale, the Antarctic surface is generally shaped by the wind (Furukawa et al., 1996). Erosion plays a promi-
nent role by scouring deposited snow, generally increasing the roughness and forming sastrugi. The deposition is also highly heterogeneous, leading to the formation of semi-organised wave-like features on the surface. Namely these are often classified in longitudinal dunes, barchan dunes, whalebacks, ripples, etc (Filhol and Sturm, 2015). The horizontal length scale of these features ranges from a few centimetres up to hundreds of meters (Frezzotti et al., 2002; Picard et al., 2014; Kochanski et al., 2018), and their height is typically 10-100 cm. Interestingly, on the Antarctic Plateau this height is generally orders of mag-
nitude larger than the averaged amount of snow accumulated during a single snowfall event, and can even be larger than the mean annual accumulation. It results that erosion can exceed accumulation in some points for some years, a situation referred to as "accumulation hiatus" (Petit et al., 1982). Annual net accumulation data obtained from stake networks at Dome C (75°S, 123°E) show indeed a wide distribution of values, including some negative ones (Picard et al., 2016). These traits are specific to the dry and windy Antarctic.

The representation of the accumulation process is usually done in a simplified way in large-scale climate models (e.g. LMDz GCM, Krinner et al. 2006) and in small-scale snow evolution models (e.g. Crocus, Vionnet et al., 2012 and SNOWPACK Lehning et al. 1999): snow is accumulated on the surface in successive layers (1D model) added at the time of the snowfall. This reflects the accumulation process as commonly observed in alpine regions. However, this representation is inadequate to account for the aforementioned Antarctic traits. Recent works tried to improve modelling with more complex deposition
schemes. Based on snowfalls recorded at Dome C Groot Zwaaftink et al. (2013) noticed that nearly half of snowfall events did not result in visible accumulation on the ground. This was explained by the fact that fresh snow is easily remobilised. They accordingly modified the SNOWPACK model, by storing snowfall in a virtual reservoir until a strong wind event triggered its release. The time elapsed in this virtual reservoir could be as long as several days. The snow was then accumulated in layers on the surface as in any 1D model and it remained permanently, neglecting erosion. (Libois et al., 2014) took a different
approach with an intent to simulate the spatial variability of snow properties observed in snowpits. They ran 50 simulations of





the 1D Crocus model in parallel. Each simulation represented a decametre-scale cell but there was neither spatial organisation nor notion of neighbourhood between the cells. The simulations were mostly independent except that they received different amount of snow during each snowfall and they exchanged snow during strong wind events (local erosion and redeposition). These two processes were implemented using stochastic rules, adjusted to mimic some in-situ observations collected at Dome

C. The approach was quite successful to reproduce the variability observed in one hundred profiles of snow density and specific surface area collected during a summer campaign. However, their rules lack a physical basis and were empirically parameterised with a limited set of observations.

    The aforementioned pioneering modelling studies are limited by the scarcity of observations providing detailed information on the accumulation process. Here, we exploit a new dataset of daily surface elevation maps obtained with an automatic

ground-based laserscanner at Dome C (Picard et al., 2016) scanning a surface area of 40–100 $\text{m}^2$ and which operated during three years. The aim is to answer three main questions: 1) what are the spatial and temporal characteristics of the accumulation patterns, 2) how long does snow remain on the surface before being eroded or definitely incorporated in the snowpack, and 3) what is the impact of the accumulation process on the snowpack's upper internal structure ? The study focuses on the meter-scale and daily to annual time scale and follows a statistical approach to describe the accumulation and erosion patterns

and their dynamics. The paper is organised as follows: Section 2 presents the data and the algorithms developed to extract information (accumulation, age of the snow on the surface, ...) from the series of elevation maps, Section 3 presents the results and Section 4 provides a discussion.

## 2   Materials and Method

### 2.1   Rugged LaserScan (RLS)

The rugged laserscan (RLS) is composed of a lasermeter mounted on a two-axis rotation stage to perform the elevation and azimuthal rotations, enabling 2-D scanning of the surface. It is described in detail in Picard et al. (2016) and only limited information is recalled here. The lasermeter (DIMETIX FLS-CH 10) measures the radial distance to the snow surface with an intrinsic accuracy of $\pm 2\,\text{mm}$ (statistical confidence level of 95.4%) which proved to be effective for a wide range of illumination and temperature conditions (Picard et al., 2016). To achieve this constant accuracy however, the rate of measurements, of 20 Hz

in optimal conditions, is automatically reduced when the conditions are unfavourable. The consequence for our particular setup where the lasermeter is rotated at a constant speed, is a reduced spatial resolution. We also found that despite a design for outdoor operations, the lasermeter performance was greatly improved in high light illumination by adding a band-pass optical filter at the laser operating wavelength (650 nm) on the optical window. The lasermeter is heated with internal components and regulated at 0°C for maximal stability of the internal time reference. We fitted an additional heating patch (20 W) on the

external box, enabling operations up to -80°C, which also contributes to the removal of the frost and snow (by sublimation) that occasionally builds up on the device.

    The two-axis stage is composed of two identical reduced motors controlled by a feedback loop on the position (servomotor). The precision and accuracy are of the order of 0.03° and 0.1° respectively, which is small but nonetheless is the main limiting





factor in terms of accuracy. The scan is performed by moving the azimuth (horizontal) stage at constant speed from nearly -90° to +90° , then by increasing the zenith angle (angle from the vertical axis) by a small increment, and finally by moving the stage back from +90° to -90°. The process is repeated many times for zenith angles from 17° up to 62°. The increment in zenith angle and the speed in azimuth are not constant, they are calculated to obtain a uniform measurement sampling over the

whole area. Nevertheless the resolution effectively obtained also depends on the actual lasermeter rate which can vary a lot. A normal scan contains about 200,000 points and takes a total of 4 hours to complete. In Picard et al. (2016) we found that the vertical accuracy is better than 1 cm on the long term (a year). The reproducibility within a scan is better than 5 mm.

Scanning is scheduled at 9 pm local time (GMT+8) every day to avoid high illumination. However, not all the scans are completed with sufficient points to produce a useful map. The main reasons include 1) downtime due to major failures of the

RLS or for maintenance, 2) snow or frost deposition on the laser window, and 3) blowing snow crystals intercepting the laser beam, which greatly reduces the acquisition rate. The two latter causes of failure obviously occur during or after snowfalls and blowing snow events. This results in fewer valid scans in the periods of greater surface change. This correlation between the observation quality and the observed phenomenon represents a potential source of statistical bias which must be kept in mind for the analysis. Other, more occasional, reasons of scan failure include power supply shutdown and interruption of the

scanning process due to undocumented errors raised by the lasermeter.

The RLS was operating at Dome C (Fig. 1) over 2 periods with different configurations (Table 1). From 1 January 2015 to 17 January 2016, it was set up at a height of 2.8 m (period 1). A major failure occurred during this period between 17 October 2105 and 5 December (49 days). After the maintenance during the summer campaign, it was reinstalled at a height of 4.5 m (period 2) on 1 February 2016. Because of the difference of height during period 1 and 2, the scanned area is different: 40 m$^2$

and 110 m$^2$ respectively, and the effective spatial resolution was accordingly adjusted to 2 cm and 3 cm respectively. Although the area scanned during period 1 was located inside the area scanned during period 2, no attempt to co-register the two was done. The two time-series are interpreted as independent datasets. Only the mean surface elevation at the end of period 1 is taken as reference for shifting the starting elevation of the period 2 in order to get a continuous mean elevation time series. Furthermore, after a year of operation in period 2, the outer sheath of the lasermeter cable got damaged by the recurring friction

during the azimuthal rotations. The electric contacts became less and less reliable. Nonetheless, the lasermeter continued to operate but returned a decreasing amount of valid data until the dismounting on 28 January 2018. Because of the random nature of the problem, a few scans are useful, at least to estimate the mean elevation of the surface. Hence, we split the period 2 into a first high quality part, named 2a, and a second part named 2b where only the few best scans are selected (Table 1). Despite a much lower temporal resolution, the period 2b time-series still provides a useful extension of nearly a year. In the following, we

mainly use the period 2a for its finer temporal resolution and wider scanned area, while the periods 1 and 2b are only exploited to investigate the general trends over the 3 years.

## 2.2   RLS data processing

Processing the raw data to produce elevation maps on a common and regular grid is performed in several steps. For each single acquisition, raw data comprise the radial distance and two angles (azimuth and zenith). After filtering the obviously erroneous





distances (less than 3 m or more than 17 m), the data are projected into Cartesian coordinates $(x, y)$ with z the vertical axis. $z(x, y)$ is the surface elevation. A second filter is applied to remove the points $(x, y)$ for which $|z(x, y) - \bar{z}| > 5$ cm, with $\bar{z}$ the averaged elevation of the points in a 5 cm-radius circle centred at $(x, y)$. This operation removes outliers and small objects like blowing crystals or the RLS mast guy lines. This set of curated but irregularly spaced points is then interpolated onto a regular

grid using the bilinear method interpolation provided by the matplotlib.mlab.griddata Python function (version 2.2). The grid spacing is set to 2 and 3 cm respectively for the periods 1 and 2, in relationship with the different setup heights. To avoid filling large gaps with the bilinear interpolation, a grid point was attributed a valid $z$ value only if at least one measurement was taken within 6 cm around it (independently of the period and grid space). If the final map contains less than 100,000 valid grid points, it is completely discarded. All these operations yield a time-series of elevation maps on a common grid. The grid

orientation was determined by observing the shadow of the RLS mast in the scanned area. We found that the x-axis lies towards 116°(East-South-East), which allowed us to draw the South direction on the maps.

Various additional datasets are then derived from the generated maps. The accumulation between every successive scan is calculated as the difference between the elevation for each point of the grid. In most cases, both scans are acquired one day apart (90% over the period 2a) so the computed accumulation is representative of the daily variation. However, due to scan

failure or rejection during the processing, longer time intervals are present in the time-series (2 days in 7% of the cases, and 3–9 days in the remaining 3%). We nevertheless interpret the accumulation time series as if it was daily accumulation only.

The age of the snow on the surface is another dataset derived from the elevation maps. The algorithm is applied independently for each point as follows. For each point, the age is initialised to 0. For each date $i$ the accumulation $a_i$ since the last detected deposition or erosion event (or the beginning of the time-series) is calculated. If $a_i$ is larger than a given positive threshold $\delta_a$,

there was deposition of new snow, and the age is reset to 0. If this amount $a_i$ is null or positive but lower than $\delta_a$, the variation is considered insignificant and the age is incremented by the time elapsed since the previous available date. If this amount is negative $a_i < 0$, snow was removed by erosion. In this latter case, the algorithm then searches back in time the first date $j$ for which the elevation was equal (or closest below) to the present elevation at the point. The snow present at the surface at the date $j$ is now emerging again. The age of this snow is calculated as its age at date $j$ plus the time elapsed since $j$, that is

$i - j$. We choose a threshold value $\delta_a = 1$ cm which tends to avoid sporadic age change for small variations (possibly due to residual noise). This algorithm is relatively robust because even if $\delta_a$ is chosen too close to the noise level, age errors are not accumulated over time, and only the statistics of the age at a given date may be affected. A low threshold tends to bias the age distribution towards younger snow due to the over-estimation of new snow accumulation. In addition to the age, we derive the time of residence on the surface, which is the same as the age except that when erosion is detected, the time on the surface at

time $i$ is taken as equal to the time on the surface at time $j$, not counting the time spent while buried $(i-j)$. From it, we also derive the time of residence before erosion, which is equal to the time spent on the surface just before erosion is detected.

## 2.3   Meteorological data

Meteorological data are used to relate surface changes and weather. We use precipitation forecast provided by the ERA Interim reanalysis (ERA-I) (Dee et al., 2011) near Dome C. Due to the absence of reliable methods to record in-situ precipitation in



Antarctic conditions, ERA-I is one of the most reliable sources of information to date (Bromwich et al., 2011; Wang et al., 2016). For wind speed and direction, we use data from the Weather station at Concordia station ( 75.1°S, 123.3°E, 3230 m a.s.l., http://www.climantartide.it/). Over the three years, the mean wind speed at 3 m is $7\,\mathrm{ms^{-1}}$ which is higher than the ERA-I prediction of $5.2\,\mathrm{ms^{-1}}$ at 10 m. Nevertheless, both sources agree on the temporal variations, with a correlation of 0.8, which is

the most important point for our comparison. The wind regime at Dome C is characterised by a prevailing direction from the South (74% of the time), the most likely direction being 190°. The distribution and maximum remain identical when selecting only strong winds (e.g. over the mean, $7\,\mathrm{ms^{-1}}$) which are relevant for this study.

## 3   Results

The RLS dataset is studied first by addressing the general characteristics of the elevation changes (Sec. 3.1) and annual accu-

mulation over the whole area (Sec. 3.2), to then focus on the smaller temporal and spatial scales (Sec. 3.3, 3.4 and 3.5). Lastly, we investigate the internal structure of the snowpack deduced from the elevation changes (Sec. 3.6).

### 3.1   Surface elevation changes

The elevation of the surface averaged over the scanned area generally increases with time at a mean rate of $8.7\,\mathrm{cmyr^{-1}}$ (Fig. 2). However, different dynamics are observed for the different years. As noted by Picard et al. (2016) for the first year, a unique

and marked accumulation event occurred in the period 4-17 July 2015 which accounted for most of the annual accumulation during that year. In contrast, the second and third years of the dataset feature a fairly regular increase of the surface elevation suggesting the absence of dramatic accumulation events. Interestingly, this regularity in the cumulated precipitation is also found in the ERA-I forecast (Fig. 2, green curve) with a remarkable absence of seasonal signal despite the huge variations of temperature between summer (typ. -25°C) and winter (typ. -70°C). However, the precipitation in ERA-I is overall too weak

with only $16\,\mathrm{kgm^{-2}yr^{-1}}$ over the three years (Petit et al., 1982). This mass is equivalent to only $5\,\mathrm{cmyr^{-1}}$ of snow taking a typical surface density value of $320\,\mathrm{kgm^{-3}}$ for the conversion (Genthon et al., 2015; Picard et al., 2014; Leduc-Leballeur et al., 2017). The missing mass in ERA-I could be due to underestimation of the synoptic precipitation or of the condensation that seems indeed very low (Genthon et al., 2015) compared to that reported by other sources (Krinner et al., 2006; Agosta et al., 2018).

The standard deviation of the surface elevation (a.k.a root mean square height of the surface) is shown in Figure 2 as a blue shaded area. Physically, this metric measures the amplitude of the elevation variations over the scanned area including all the spatial scales of variations, that is those due to the meter-scale roughnesses, the local slope and potential instrumental artefacts (such as the repeatability error of the RLS). Overall, the standard deviation has a large magnitude compared to the annual accumulation. Over the three years, the standard deviation was smaller (4 cm) at the beginning of the first season,

before the event in July 2015, where it nearly doubled and remained fairly constant after that to 7 cm and then 8 cm from June 2016. In particular it remained constant during the shift period 1 and 2a, despite a three-fold increase of the scanned area. The increases in the last period (2b) seems unreliable as they are most probably explained by the failures of the RLS. To estimate the



respective role of the meter-scale roughness and the local slope over the scanned area, we fitted a plane on each scan using the least square method. The local slope is fairly constant, of the order of 1–1.5° over the time-series, indicating that a large-scale terrain undulation was present. In comparison, the Dome C area has a very small overall slope, less than 1 m per kilometer (0.06°). Once the plane is subtracted from the elevation map, the standard deviation is reduced by about half, implying that half

of the elevation variations are attributed to meter-scale roughness (typically sastrugi and small dunes) whilst the other half are caused by terrain undulation (e.g. decameter-scale dune Picard et al. (2014)). It is finally worth noting the absence of seasonal signals in the roughness evolution, which differs from reports for other locations on the ice-sheet (Gow, 1969; Adodo et al., 2018).

## 3.2 Spatial characteristics of the annual accumulation

Figure 3 shows the statistical distribution of annual accumulation over the scanned area. The three periods, corresponding roughly to the years 2015, 2016 and 2017, present different patterns. In 2015, the mean accumulation is 7.7 cm and the highest accumulation is nearly 30 cm. Negative accumulation (called hiatus) concerns 12% of the surface. The second year highlights a significantly higher mean accumulation with 10.0 cm (+31% compared to 2015) which is also present in ERA-I (+50% precipitation). It results that almost no negative accumulation is observed, but more surprisingly the maximum accumulation

is reduced compared to 2015, to about 22 cm from 30 cm. The accumulation distribution in 2016 looks Gaussian and narrow in contrast to the two other years showing a wider and more asymmetrical distribution. The last year has the same mean accumulation as the first one, but less negative accumulation values and a lower maximum accumulation, of the order of 22 cm.

In Picard et al. (2016), it was noticed that the distribution of annual accumulation estimated over the RLS area of 40 m² in

2015 was surprisingly similar to the distribution obtained from the GLACIOCLIM stake network which is composed of 50 measurement points distant of 10 m, thus covering a much more extensive area. Figure 3 further confirms this finding for the two last years (2016 and 2017). This remarkable result indicates that despite the small extent of the RLS scanned area, the distribution of net annual accumulation is representative of a wide area.

## 3.3 Spatial and temporal characteristics of the daily accumulation

The mean and standard deviation of accumulation (noted $\bar{A}$ and $\sigma_A$ respectively) between every two consecutive dates (called daily accumulation here despite the irregular sampling in time) were calculated and compared to precipitation (from ERA-I) and wind speed and direction (from the Concordia Automatic Weather Station). Figure 4 shows scatter plots between these variables (except wind direction as no interesting signal was found). Overall, there is little correlation, if any, between the variables, which is unexpected because in many regions of the world the daily accumulation is strongly related to precipitation

(e.g. Alpine regions), and in Antarctica the role of the wind has been emphasised (Groot Zwaaftink et al., 2013). This role may be the reason why in the bottom left plot in Figure 4, points are scarce above the orange line delineating a domain with low-wind / high surface change. This indicates that some wind is necessary to induce surface change, which is obvious, though the statistical relationship is probably not robust. Another interesting and more robust pattern is visible in the bottom right




plot showing the mean and standard deviation of the accumulation. This pattern indicates that significant (positive or negative) mean accumulations are always associated with high standard deviations over the area, which can be expressed mathematical by $\sigma_A > |\bar{A}|$. We interpret this relationship by the fact that 1) many events change the surface but do not affect the overall mass in the area ($\sigma_A >> |\bar{A}| \approx 0\,\mathrm{cm}$) and 2) that the events leading to significant accumulations or erosions cause highly

heterogeneous changes over the area (with both negative and positive changes), and conversely that the process of accumulation or erosion by "layer" (which would correspond to $\sigma_A << |\bar{A}|$) does not exists at Dome C. However this relationship does not link accumulation to meteorological conditions.

This result suggests that further investigation concerning the positive and negative changes of surface elevation in every point is necessary. Figure 5 shows the cumulated sum (average over the scanned area) of daily accumulation considering either only

the increments (deposition above 0, 0.5 or 2 cm) or the decrements (erosion higher than 0, 0.5 and 2 cm). The figure also shows the net cumulative accumulation for comparison, which is exactly the same as the mean surface elevation changes in Figure 2. Over the one-year period, each pixel has received a total of 73 cm of snow on average. Most of it has been removed (63 cm, or 87%) leaving a final net accumulation of about 10 cm (Section 3.1). The ratio between deposition and net accumulation suggests that a snow particle is remobilised at least $73/10 = 7.3$ times before its definitive settling. This value is to be taken

with caution, firstly because of the limited frequency of the scans that only allow to probe the "long term" deposition/removal cycles, which excludes the rapid rebounds that occur during saltation and blowing snow, and secondly because of the limited precision of the laserscanner that leads to count any small measurement errors as positive or negative accumulation. For this latter reason, we additionally computed the increments above 0.5 cm (this value is twice larger than the RLS precision). These "significant" events of deposition bring 56 cm of snow in a year which is still much larger than the net annual accumulation and

again suggest that snow is subject to many deposition/removal cycles. Interestingly, the increments above 2 cm (call hereinafter "major events" as they represent more than a 5th of the annual net accumulation) amount to 22 cm, meaning that every pixel receives many times a year at least 2 cm of snow in a single day (or a few days for the few cases the scanner was not working). The process of erosion shows similar behaviour, with 18.4 cm removed by events larger than 2 cm on average over the area.

Overall, the results of this section depict a surface subject to frequent changes that remains invisible to the observers who

only access long time-period averages or spatial averages of accumulation. To further explore the accumulation process, in the next section we focus on the "major events" which account for 30% of the total deposition.

### 3.4 Patchy accumulation

Figure 6 shows the accumulation map between 4 and 17 July 2015 corresponding to the exceptional event rising the surface by 17 cm on average and clearly visible in Figure 2. Unfortunately no data is available between these two dates, preventing to

describe the precise sequence of this event. This RLS failure was certainly caused by snow jamming the lasermeter window, which leads to suppose that intense snow drift started at the beginning of this period, right after 4 July 2015. This event is the only one rising the surface as a whole and smoothly. Nonetheless, the accumulation is not even, instead it features a trend with increasing $x$ from nearly 0 cm (left corner in the figure) to 40 cm over a distance of about 10 m (right corner). This corresponds to the decameter-scale dune invoked above.





Figure 7 shows very different accumulation maps, corresponding to the accumulation between 28 and 29 May 2016 and between 29 and 30 May 2016. It is worth noting that the x- and y-axis scales and the colorscale are different from those in Figure 6. The first day, the net accumulation is -0.8 cm, with 80% of the area in erosion (and 75% in large erosion (>2 cm)). The second day, the accumulation is +0.3 cm with 62% of the area in accumulation. In both cases, elongated patterns are

clearly visible. The second day it is possible to observe that most erosion patterns correspond to the accumulation patterns of the previous day (e.g. the two large blue areas on the left the first day are green the second day) meaning that most of the accumulation features in the first day have been blown away and replaced by new ones with similar shapes (size, elongation, orientation).

To explore the geometrical characteristics of these deposition patterns, that we called patches, we applied a threshold (2 cm)

to the daily accumulation maps and segmented the map. Each individual patch received then a unique label and its geometrical properties were computed using the Python function skimage.measure.regionprops. The properties include area $A$, eccentricity $e$ and major axis length of the best-fitting ellipse (0 indicates a circle and 1 an infinitely-thin segment), and the orientation of the major axis (with respect to North, increasing Eastward). Figure 8 reports the orientation and eccentricity of the elongated patterns only ($e > 0.8$). On 29 May 2016, it is clear that the patches were aligned with orientations ranging from 140° to 192°

with respect to the North. The average orientation is 174° (we applied weighting proportional to $Ae^2$ to account for the size and eccentricity). This orientation is nearly South which unsurprisingly corresponds to mean wind direction of the day (169°) and to the prevailing wind direction (Champollion et al., 2013). The same computation is repeated for the 280 days of the period 2a, yielding 1103 patches that appeared for 93 different days (33% of the time). As found before for the mean accumulation, there is no clear relationship between the patch properties and wind speed (not shown). The most marked relationship is between the

daily-mean orientation and the wind direction on the same day (Fig. 9) with a correlation of 0.33 (the correlation is weighted by the number of patches as in Fig. 9). This simply means that the patches form longitudinal dunes or sastrugi approximatively aligned with the wind direction.

As a consequence of the frequent deposition/removal cycles, many patches studied here are in fact ephemeral. They can be rapidly removed by the next wind event following their formation, as highlighted on the sequence of 29 and 30 May 2016.

These ephemeral patches do not contribute to the snowpack on the long term. To capture and investigate more specifically the few patches that remain and settle and that in the end are the important ones for the snow mass balance and the snowpack internal structure, we estimated the time of survival as follows: for each patch detected at time $i$, we compute its initial volume (between the surface at time $i$ and the surface at time $i - 1$) and how this volume evolves at all times $i' > i$ (volume between the surface at time $i'$ and the surface at time $i - 1$). More precisely we seek for the minimum value of this volume, and the date

at which this minimum is reached. The patch is fully eroded if the minimum volume is $0\,\mathrm{cm}^3$ or negative, which happens for 652 of the patches among 1103 detected in the period 2a (59% of the cases). Partial erosion occurs when the minimal volume is between 0 and the initial volume, which concerns all the other patches. We found none being fully and continuously buried after deposition in our dataset. Full or partial erosion of at least half of a patch volume occurs for a large proportion of them (930, i.e. 84%). The mean time of survival before full erosion is 17 days, and for partial erosion this time increases to 46 days,

meaning that partial erosion is typically possible after a longer period. In other terms, if a patch is not removed soon after





deposition, it is increasingly harder to remove it, as expected because of sintering. The calculation of the time of residence before erosion (Section 2.2) provides similar information while not being limited to patches (accumulation over 2 cm). The time of residence on the surface before erosion is 15 days on average over the period 2a, with a large spread (10 days standard variation). The maximum is 129 days.

This result confirms that deposition/removal of snow is a very common process and that only a small part of the deposited snow remains on the surface and contributes to the snowpack. Moreover, even when the snow is removed, the number of days spent on the surface can be significant, which lets metamorphism operate and modifies the physical and chemical properties of the snow before it is removed and blown away.

### 3.5    Age of the snow on the surface

We estimate the age of the snow on the surface with the algorithm described in Section 2.2 for all the scans. Figures 10 and 11 show the age of snow near the end of the period 2a (18 and 27 January 2017) with respect to the beginning. We have chosen these dates because they are less affected by the fault that increasingly affected RLS from February 2017.

The map and the distribution of age show that snows with very different ages are present on the surface, from 0 (accumulation of the day) to the maximum possible given the duration of the time-series. About 55% of the surface is younger than 100 days
(which is thus the approximate median age), while about 18% is older than 200 days and 5% older than 300 days. These proportions change from day to day because of the ephemeral deposition/erosion as illustrated on 27 January 2017 (Fig. 10) where a few patches of recent accumulation covered 11% of the surface. These patches disappeared the day after (data not shown), forming the surface as shown in the 18 January map. It is also clear from the histograms that almost all ages are present at the same time, which is in agreement with the apparent regularity of the accumulation (Fig. 3).

Note that period 2a had a relatively high net accumulation and narrow spatial distribution of accumulation, so it is likely that the distribution of age is even wider for years with less accumulation.

The age maps provide another interesting piece of information. We can indeed see clear patterns with marked alignments on these maps (Fig. 10). This confirms the patchy nature of the accumulation process already noted in the previous section (Sec. 3.4). We indeed showed that a significant part of the daily accumulation over 2 cm is organised in patches but that most of them
(59%) do not contribute at all on the long term to the surface mass balance. Here, we complement this by showing that some of the partially eroded patches remain clearly visible on the surface even after a year.

### 3.6    Structure of the snowpack

The accumulation process depicted in the previous section may affect the internal structure of the snowpack and its spatial variability. The burial of the snow can be tracked using the RLS, at least over a short depth given the limited length of the
time-series. Figure 12 shows the snowpack internal structure along the x-axis transect at $y = 4\,\mathrm{m}$ (see e.g. Fig. 10) at the end of period 2a. It is obtained, assuming no compaction, by plotting the successive positive increments of surface elevation in each point with a colour depending on the date of deposition. The grey colour marks snow older than the first day of the period 2a





(1 Feb 2016). The $z$ origin corresponds to the mean elevation in the whole scanned area at the beginning of period 1. The two grey lines represent the mean at the beginning and end of the period 2a respectively.

It is remarkable that the figure shows distinct coherent patterns and fairly little noise. The figure confirms that the heterogeneity on the surface transfers into the snowpack. For instance, around $x = 0\,\mathrm{m}$ old snow is present at $11\,\mathrm{cm}$ under the surface (yellow) whilst the surface is young (black). This area presents the greatest diversity in terms of age and the largest number of distinct layers. Around $x = -3\,\mathrm{m}$ in contrast, the whole accessible depth features a single homogeneous layer formed in March 2016 (orange). Another thick homogeneous layer is found around $x = -7.5\,\mathrm{m}$ but it is much younger (violet, August 2016). Even where we do not observe a unique layer, it is clear that only a few events ($\approx 5$) have contributed to the snowpack, forming a few distinct layers. Conversely, it means that in every point many deposition events occurring during a year are not represented, which implies that the profiles of snow physical and chemical properties are probably very different from a point to another.

Another remarkable feature is around $x = 7.5\,\mathrm{m}$ where we can see a $10\,\mathrm{cm}$ high dune deposited in July 2016 (orange) where the surface was already higher than the surrounding as marked by the grey lines. Then, two successive events (in September and November, violet) accumulated more snow on the sides of this dune which was then about $15\,\mathrm{cm}$ higher than the surrounding surface. This may be the result of interaction with the pre-existing dune leading to preferential deposition on a side, here the windward side. A similar behaviour is observed around $x = -4\,\mathrm{m}$ where the space between the two small dunes (in light orange) has been filled by some subsequent events.

## 4 Discussion

The laserscanner that was operating at Dome C for about three years provides very rich and new information on the accumulation process. We analyse these results successively from a temporal, spatial and snowpack perspective.

### 4.1 Temporal perspective

The RLS provides contrasted results regarding the surface evolution depending on the spatial scale of interest. On the one hand, the time-series of surface elevation averaged over the whole scanned area depicts a slow and relatively regular accumulation, without seasonality or rapid events, except a single event which raised the surface by $8\,\mathrm{cm}$ in a few days during the first winter of observation. The accumulation rate measured by RLS ranges between 8 and $10\,\mathrm{cm\,yr^{-1}}$ for the three years of observation, which agrees with values reported by previous studies for Dome C (e.g. Petit et al., 1982; Urbini et al., 2008). The time-series of standard deviation of the surface elevation (RMS height) also is relatively steady, which again suggests that only slow changes occur over the scanned area. This overall stability is further illustrated in the sequence of photographies in Figure 13. The photographies were taken from $20\,\mathrm{m}$ above ground at Dome C. They depict a landscape dominated by barchan dunes with little differences between the pictures of January 2017 and December 2017, 11 months apart. This steadiness is surprising because the accumulation over this period has been of the order of $8\,\mathrm{cm}$, the typical annual accumulation expected in the area (Petit et al., 1982; Genthon et al., 2015).





On the other hand, a very different picture is obtained by investigating the daily changes of surface elevation (i.e. the daily accumulation) and the small spatial scales. Indeed, many local changes of elevation affect the surface almost everyday. Most of these changes are however ephemeral (a few tens of days) so that the overall statistics of the surface are relatively constant or slowly changing. These changes are typically caused by migrating patches on their way windwards or remobilisation of

loose snow deposited by a recent storm. The picture of September 2017 in Figure 13 illustrates how major these ephemeral changes of the landscape can be. Nevertheless, these temporarily accumulated snow masses have little consequence as far as the surface mass balance and the snowpack internal structure are concerned. They however do have consequences on other aspects as they shield the snow surface for a few days, weeks or even months from solar and infrared radiation and suppress the photochemistry activity and the exchanges of heat, water and chemical components between the consolidated surface

and the atmosphere. During the period of residence, these ephemeral snow masses are subject to transformations resulting in changes of their physical, isotopic and chemical properties (Casado et al., 2018). When these masses are transported in a downwind region and deposited, they can be very different compared to fresh snow coming from direct local snowfall. This phenomenon of deposition/erosion/transport repeats itself several times. We provide a rough estimate of about 7 cycles before settling, considering only the timescales longer than a day, thus excluding the many rebounds occurring during the saltation.

We however are unable to estimate the distance traveled over these cycles.

The meteorological conditions triggering the surface changes is an important question for modelling, but has not been elucidated here. Snowfalls seem to be frequent at Dome C according to the ERA-I reanalysis, which is in agreement with the slow and regular accumulation observed with RLS, but these events are not related to the observed changes of the surface. The occurrence of snowfalls in ERA-I has been compared to satellite data (Palerme et al., 2014; Lemonnier et al., 2018) in a coastal

region and found to be reliable, but on the other hand, it is well known that ERA-I misses a large part of the accumulation amount around Dome C (Genthon et al., 2015). Apart from snowfall, common sense suggests that wind speed should be a driver of change. Groot Zwaaftink et al. (2013) considered in a modelling study that snow deposition occurred after sustained wind over $3\,\mathrm{ms}^{-1}$ for 100h. Libois et al. (2014) similarly considered that wind speed over $7\,\mathrm{ms}^{-1}$ is required to initiate drift which then continues as long as this speed limit is reached at least once within 24h. This criteria yields a drift rate of 70 events

per year, which compares well with our estimates of 93 days with patch formation over a year. Nevertheless, our analysis has not revealed any robust simple relationship with the wind speed. The accuracy of the daily accumulation derived from RLS is limited by noise and some artefacts, which could be an explanation. However, a complex interaction between wind and the surface is also not to be excluded. For instance, wind with a direction perpendicular to the prevailing surface roughness exerts a stronger drag on the surface than when parallel, potentially resulting in stronger erosion. This effect was at least once shown on

the removal of surface hoar at Dome C (Champollion et al., 2013). Similarly (Amory et al., 2016) showed how drag decreased during a snowfall episode (though not at Dome C, but in a coastal region) as the surface roughness direction was adjusting to the wind direction. Another key parameter, here missing in our analysis, is the cohesion of the snow (Sommer et al., 2017) that is able to modulate to a very large extent the snow mobility (Vionnet et al., 2012). The cohesion is due to sintering and increases over time at a rate increasing with temperature. Hence, time elapsed and temperature conditions since the initial

deposition could be relevant parameters to be included in a relationship between wind speed and surface changes. However,

we found that many patches of accumulation are removed within a few days only, which is probably too short for sintering to be effective (Sommer et al., 2018).

## 4.2 Spatial perspective

All our results show great variability at the meter scale on the surface (e.g. daily accumulation, age of snow on the surface) and within the snowpack (snow layers). Most of this variability results from the heterogeneous accumulation of snow and the selective erosion. We propose to call this former process "patchy" accumulation, in contrast to the even accumulation "in layers" which is typical of the alpine regions. Representing this process in one-dimensional snow models is a challenge. (Groot Zwaaftink et al., 2013) managed to represent the fact that most patches are ephemeral by summing snowfalls and delaying the effective deposition based on wind speed based criteria as aforementioned. Even though this results in thicker deposition per event than when all snowfalls are deposited independently, it is impossible with one-dimensional models to account for the fact that the patches often cover a very small fraction of the surface and therefore can be much thicker than if a snowfall is evenly deposited. This is fundamental to produce layer thickness of a few centimetres, as observed in the field, when most snowfalls bring less than a millimeter per day (4 mm is the maximum of snow over the three year period in ERA-I). Only using a distributed or three-dimensional approach enables the representation of this feature. Libois et al. (2014) attempts (already mentioned) were a relative success with a good representation of the spatial distribution of annual accumulation compared to the GLACIOCLIM stake network. However, some of their hypotheses need to be reevaluated in the light of our new results. For instance, they assumed that snow deposition only occurs in the 20% lowest part of the surface. This tends to smooth the surface, yet we have shown an example of accumulation near the highest dunes, which conversely tends to enhance the spatial variability. This may be the reason why they obtained an under-estimation of the variability of the density and specific surface area profiles in the snowpack. Further work is needed to implement a process able to produce dunes and more generally to transfer the observational findings of the present study to concrete processing, adequate for numerical modelling.

It results from the patchy accumulation and the strong erosion that the surface is continuously rough at Dome C, as evidenced by the standard deviation of surface elevation. This is usually not the case in alpine regions where roughness increases after snowfalls (Naaim-Bouvet et al., 2016). Surface roughness can amplify itself, first because it plays the role of aerodynamic obstacles that promote heterogeneous deposition and the formation of new rough features overlying the old ones. Moreover, on a longer term, snow on the different faces of the roughness features is exposed to different radiation and wind shear conditions, likely leading to different evolutions of the microstructure (different sintering, sublimation, condensation and metamorphism). The RLS is limited on this aspect. An avenue is to exploit the laser backscatter signal available from some lasermeters, to retrieve the specific surface area, micro-roughness (hoar), cohesion and potentially other properties.

It is also worth mentioning that the patches studied throughout this paper are smaller in general than the gradient of accumulation that appeared in July 2015 (Fig. 6) and the barchan dunes visible in the photographies in Figure 13. The former are also well aligned with the wind direction (longitudinal bedform) while the latter have tails elongated at ≈45° with respect to the wind direction (Filhol and Sturm, 2015). These clearly are different objects, with different size (meter versus decameter) and different dynamics (daily versus yearly).





### 4.3 Snowpack perspective

The heterogeneity of the surface eventually transfers to the snowpack in depth. Figure 14 shows a vertical section of snow extracted from 5 m depth with evidence of past windpacks. Several layers appear distinctively despite the age (over 50 years old). It is not excluded that transformations of the snow after burial may have amplified the initial differences of snow properties,

but in any case, these layers are thick, and were certainly thick when deposited, compared to the annual accumulation. These layers are maybe even thicker than most patches identified with RLS or in the snowpack reconstruction (Fig. 12). An important consequence of this internal heterogeneity concerns the interpretation of ice cores at high resolution or measurements along profiles. With snow age on the surface spanning at least one year, it is clear that some precipitation events, volcanic eruptions, or deposition of nuclear substances are not recorded everywhere, at least not with the same intensity. Gautier et al. (2016)

explored in detail this aspect for volcano traces using five cores extracted one meter apart at Dome C. They found that volcanic events were missing in 30% of the cores on average and that the flux uncertainty reached 65% when a single core was used. Another issue is the variable age of the snow at a given depth. We showed variations up to one year, but it was likely under-estimated due to the limited duration of our time-series. This age spread not only hinders the analysis of any annual and sub-annual signals, but also may explain some apparent multi-year signals as discussed by Laepple et al. (2018). This also

suggests that depth synchronisation should be applied between different cores. Gautier et al. (2016) found a maximal offset of 40 cm between two of their five cores, which is the high-end of what can be explained with our estimates of standard deviation of surface elevation (up to 8 cm) or annual accumulation range (up to 30 cm).

### 5 Conclusions

The laserscan dataset collected at Dome C over three years provides for the first time quantitative information on the snow

surface dynamics in a site typical of the ridge area on the East Antarctic Plateau. The main results demonstrate that i) the surface elevation increases on average with an apparent regularity, without seasonality, iii) the variations at meter-scales are in contrast large and highly dynamical, iii) the surface is continuously rough, with a standard deviation of up to 8 cm, iv) the accumulation and erosion events are frequent, spatially uneven and significant with respect to the annual net accumulation which implies that snow is remobilised several times before settling, v) the age distribution of snow on the surface spans

over more than a year, vi) the snowpack internal structure reflects the surface heterogeneity. These results are useful and have great consequences for several research topics including surface mass balance, surface energy budget and thus climate, photochemistry, snowpack evolution and signals archived in ice cores, which requires further work. We also plan to improve the RLS to capture smaller time scales (hourly) and attempt to increase its robustness in order to collect longer time-series as this proved to be important to assess the age of surface snow and capture the inter-annual climate variability. The present

results can also be exploited to build stochastic or physical modelling of the accumulation and erosion process. Investigating other locations on the Antarctic plateau, with different annual accumulation and wind speed is necessary.





*Code and data availability.* The RLS dataset and the code to produce the figures will be made available after the review process.

*Author contributions.* L. Arnaud and G. Picard developed the Rugged LaserScan (RLS) which was deployed and maintained by E. Lefebvre at Dome C. R. Caneill conducted a first analysis during his master thesis, under the supervision of G. Picard, L. Arnaud and M. Lamare. The analysis was extended by G. Picard. All authors contributed to the manuscript.

5    *Competing interests.* The authors declare no competing interests.

*Acknowledgements.* The RLS was developed under the ANR program 1-JS56-005-01 MONISNOW and by a grant from OSUG@2020 (investissement d'avenir – ANR10 LABX56). The authors acknowledge the French Polar Institute (IPEV) for the financial and logistic support at Concordia station in Antarctica through the NIVO program. We acknowledge V. Favier for providing the GLACIOCLIM stake network observations. In-situ metereological data and information were obtained from IPEV/PNRA Project 'Routin Meteorological Observation at
10   Station Concordia' - http://www.climantartide.it. We also would like to thank Florent Domine for his helpful comments".



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





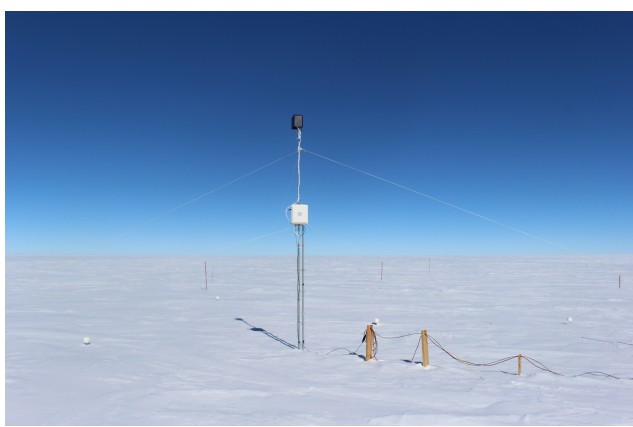

**Figure 1.** The RLS setup at Dome C (January 2017). The lasermeter and rotation mount are located under the black cap. The control is in the white box. The photography looks southward, facing prevailing winds and the scanned area is behind the RLS mast.

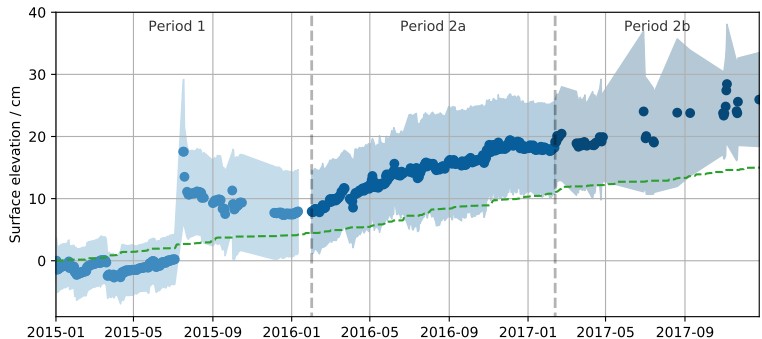

**Figure 2.** Evolution of the mean and standard deviation of surface elevation in the scanned area observed by RLS (blue) and of the cumulative precipitation forcast from ERA Interim at Dome C (green) converted in snow depth assuming a surface density of $320\,\mathrm{kg\,m^{-3}}$. The blue shade shows $\pm 1$ standard deviation around the mean.

**Table 1.** RLS configuration and performance for different periods.

| Period name | Dates | Height (scanned area) | % Success |
|---|---|---|---|
| 1 | 1 Jan 2015 to 17 Jan 2016 | $2.8\,\mathrm{m}$ ($40\,\mathrm{m^2}$) | 65% |
| 2a | 1 Feb 2016 to 11 Feb 2017 | $4.5\,\mathrm{m}$ ($110\,\mathrm{m^2}$) | 79% |
| 2b | 12 Feb 2017 to 25 Dec 2017 | $4.5\,\mathrm{m}$ ($110\,\mathrm{m^2}$) | 16% |





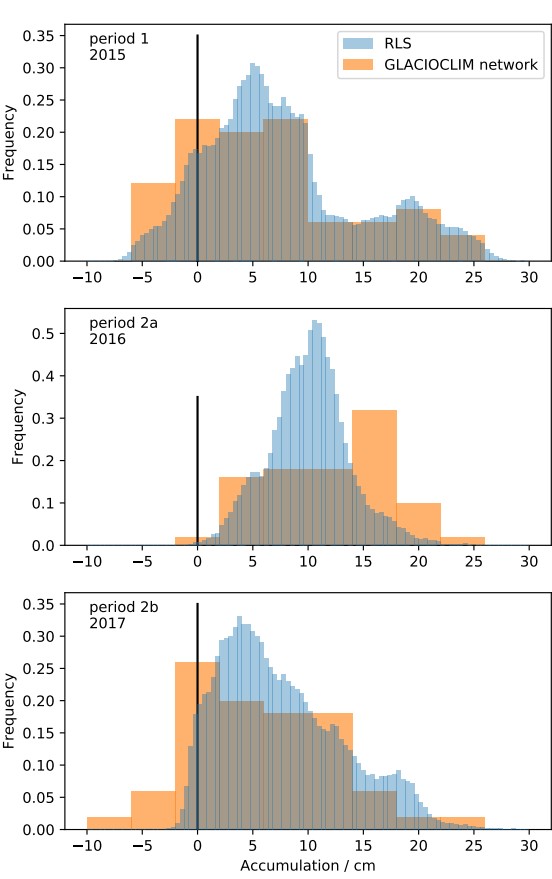

**Figure 3.** Distribution of the annual accumulation in the scanned area and from the 50 stakes of the GLACIOCLIM network near Dome C.





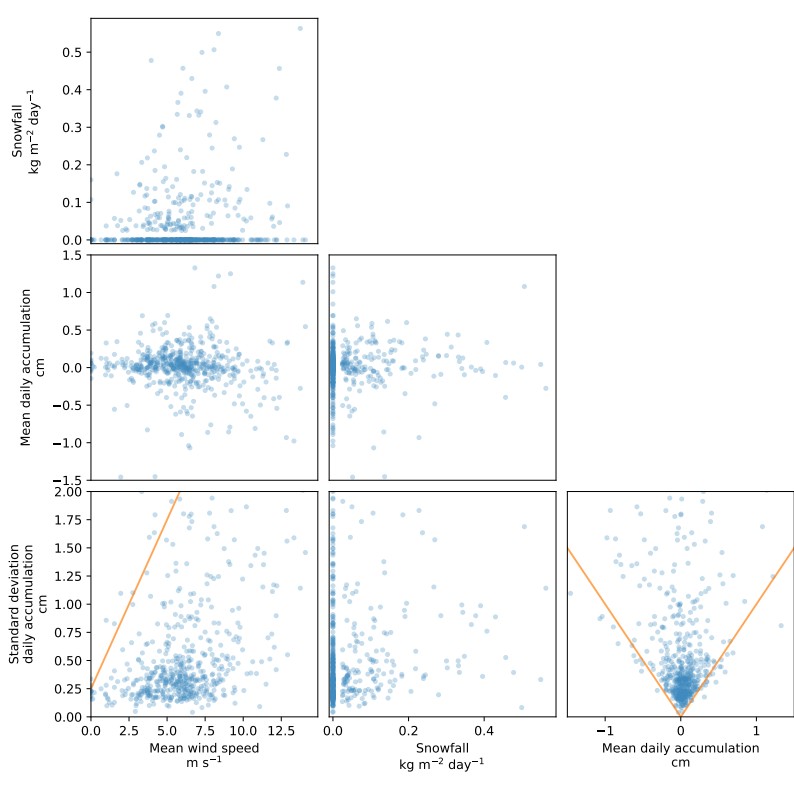

**Figure 4.** Comparison between mean daily accumulation, standard deviation of daily accumulation, daily snowfall, and mean daily wind speed including data from the three years. The orange lines delineate remarkable zones with little data (see Section 3.3).





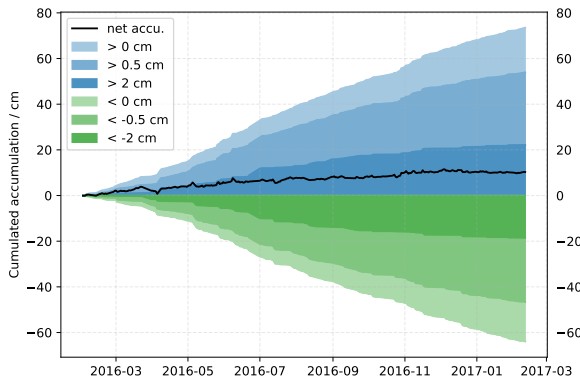

**Figure 5.** Cumulative spatially averaged amount of snow accumulated (blue $> 0$) and eroded (green $< 0$) in every pixel during the period Feb. 2016 - Feb 2017 (period 2a). Only events with accumulation over the threshold $0.5\,\text{cm}$ and $2\,\text{cm}$ and erosion below the thresholds $-0.5\,\text{cm}$ and $-2\,\text{cm}$ are taken into account for the lighter blue/green curves. Cumulative net accumulation is shown in black.

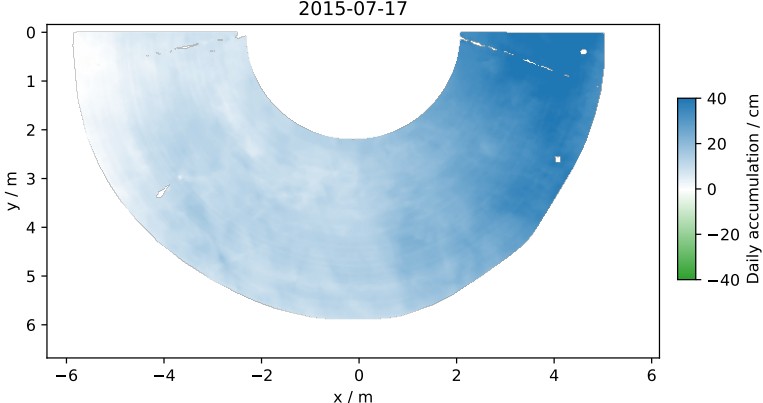

**Figure 6.** Accumulation between 4 and 17 July 2015.





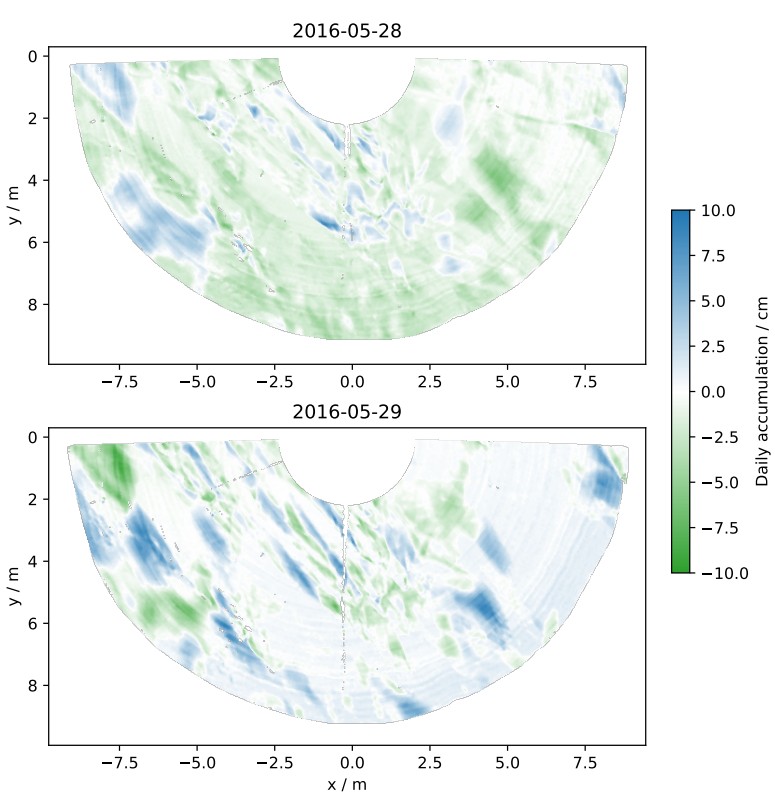

**Figure 7.** Accumulation between 28 and 29 May 2016 (top) and 29 and 30 May 2016 (bottom).





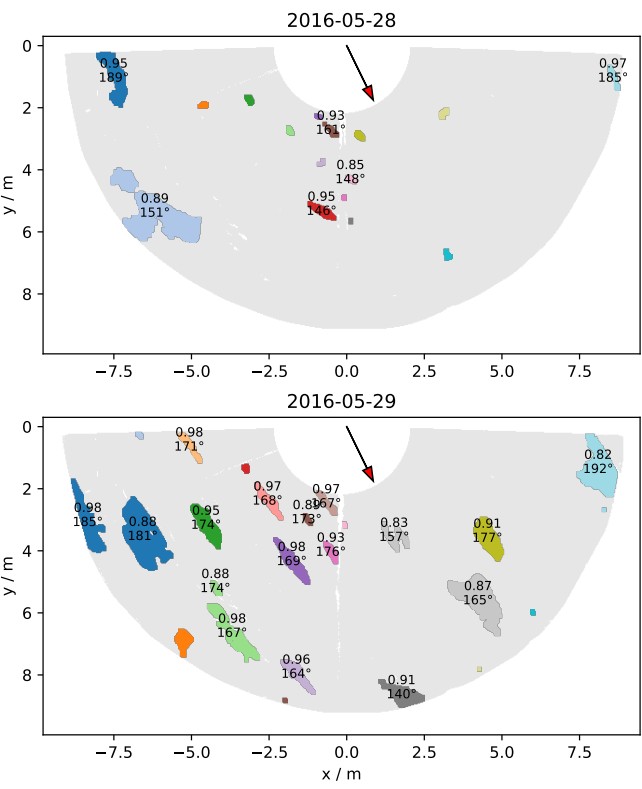

**Figure 8.** Patches of accumulation (>2 cm) between 28 and 29 May 2016 (top) and 29 and 30 May 2016 (bottom). For each patch with sufficient elongation, the circularity and the orientation is indicated. The South direction (180°) is indicated by the red arrow.





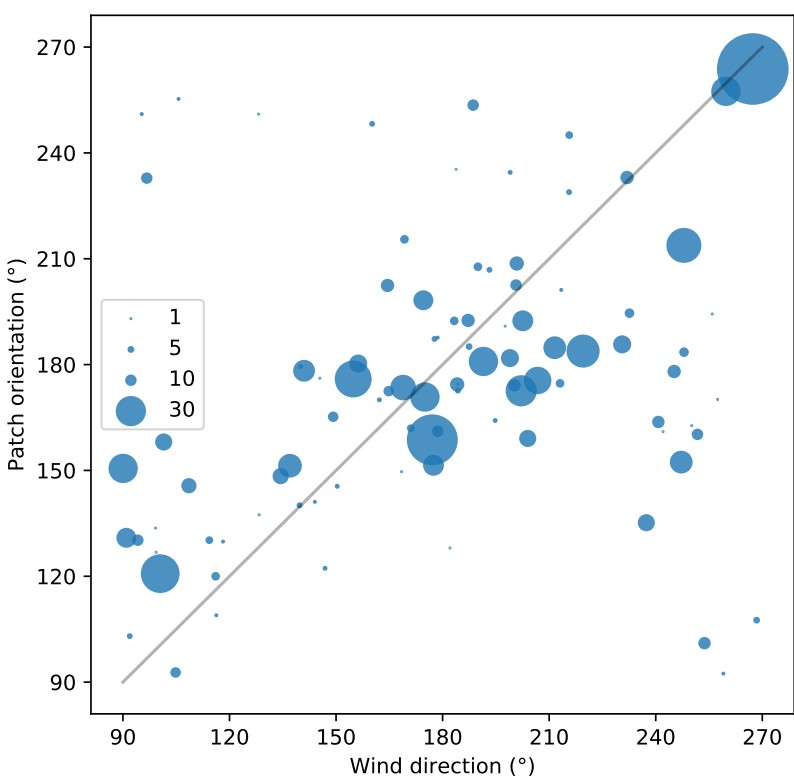

**Figure 9.** Daily-mean orientation of the accumulation patches as a function of the daily-mean wind speed. The size of the symbols indicates the number of patches.



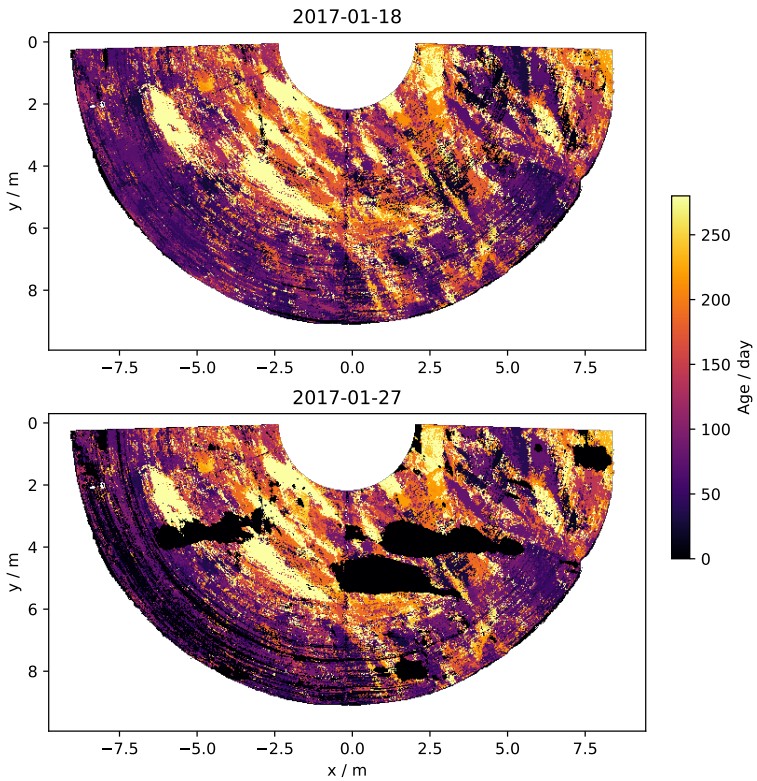

**Figure 10.** Map of age of the snow on the surface for two dates.

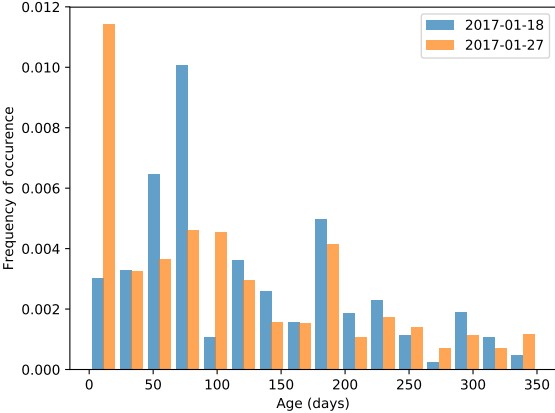

**Figure 11.** Distribution of age of the snow on the surface for two dates.



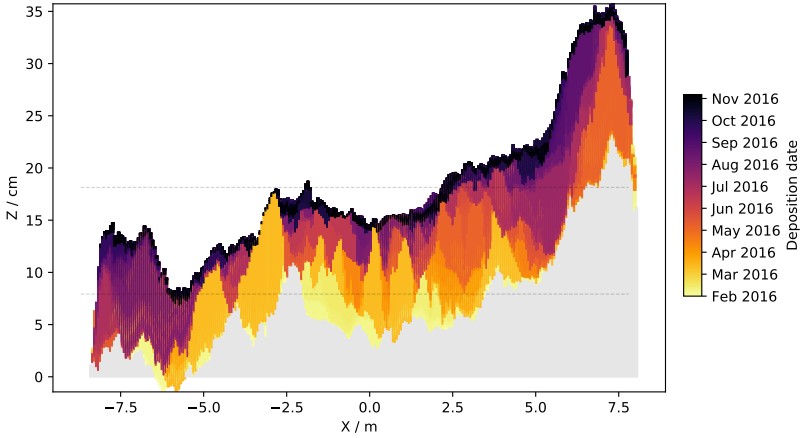

**Figure 12.** Snowpack internal structure along the transect parallel to the X-axis at $Y = 4\,\mathrm{m}$ deduced over the period Feb. 2016–Feb. 2017 (period 2a). The color indicates the date of deposition of the layer. The gray shade corresponds to snow older than the first acquisition of RLS in this period. The two gray lines represents the scanned-area average surface elevation at the beginning and end of the period 2a.

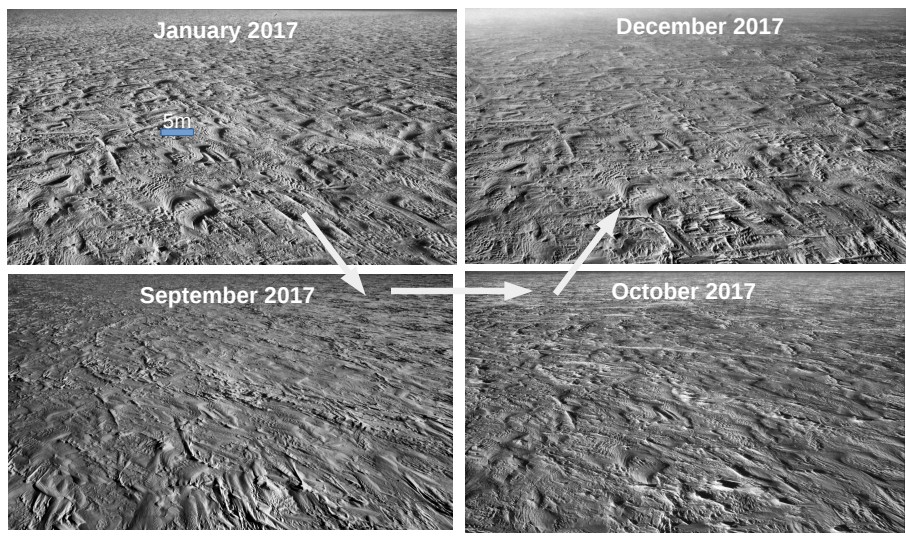

**Figure 13.** Photographs in the near infraread ($820\,\mathrm{nm}$) taken from $20\,\mathrm{m}$ height at Dome C ($75°\mathrm{S}$, $123°\mathrm{E}$) in Antarctica on 27 Jan. 2017 (9pm), 21 Sep. 2017 (6am), 23 Oct. 2017 (8pm), 4 Dec. 2017 (5pm) and highlighting how little the surface can have changed over nearly one year.





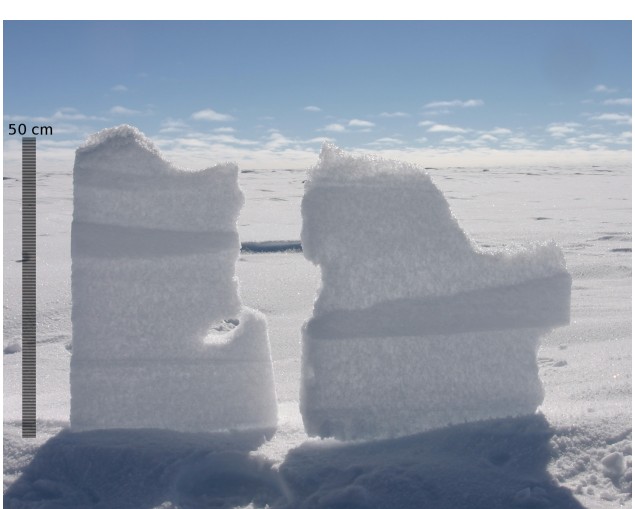

**Figure 14.** Photograph of thin vertical section extracted from 5 m depth at Dome C in January 2010 showing the persistence of heterogeneity at depth.