# Peer review of "Observation of the process of snow accumulation on the Antarctic Plateau by time lapse laserscanning"

_The Cryosphere, 2019_

## Referee Comment (RC1) · Anonymous Referee #1 · 29 Jan 2019

The authors show a very interesting laser scanner dataset capturing the patchy accumulation of snow at Dome C. The analysis and discussion correctly highlight the issues this accumulation pattern may create for interpretation of presumed horizontal stratigraphy. The paper is of high quality and merits publication after a few issues are fixed. 1. Is the major accumulation event real or did the scanner tower tip. Convince your reader/reviewer of your conclusion. 2. Errors of the scanner precision under real world applications and errors added by interpolation process are underrepresented in the analysis and not used as benchmarks in deciding when erosion or accumulation is detected. They could be used to argue for statistical certainty of erosion or accumulation. 3. Some more thought should be given to the role of repeated redistribution

in homogenization of the snowpack. The authors present the patchy accumulation as if it assures signals from short lived events (e.g. eruption ash) will be absent from many locations. It is also true that the redistribution vertically mixes short lived events, however, and may after many redistributions, place evidence of them at most locations.

Minor corrections follow: Page 2 Line 3 – "model's skills" check – did you mean model skill? models' skill Line 5 – condensation is transition to liquid from vapor. Deposition is phase transition to solid from vapor. Deposition is likely what is meant. Line 7 – maybe clarify here that sublimation of blowing snow possible too, not just "from the surface"? Line 14 – . . . scale of interest get(s). . . Line 16-18 – It is not clear whether erosion or deposition is the predominant cause of roughness, or which increases the roughness more. Over time, the authors note that the roughness remains nearly constant. Perhaps remove the statement that erosion is "generally increasing the roughness"

Page 3 Line 7 The approach was quite successful (in reproducing) the. . . Line 15 delete space before ? Line 29 consider clarifying . . .. "in high (natural) light illumination" Line 32 ". . . enabling operations (at temperatures as low as -80C). . ."

Page 4 Line 9-11 The criterial of 6 cm here, combined with an interpolation to 2 and 3 cm grids, respectively could be generating a considerable amount of 'made up' data. Can you defend this better for your reader? How many points did not have an actual observation within 1 or 1.5cm radius (the interpolated point spacing)? What is the total number of points before and after interpolation, particularly in low density areas of the survey?

Page 5 line 20-30 It is unclear in this discussion if the threshold for positive change is also applied to negative change. The discussion also does not incorporate any handling of instrument surface position detection error, which sounded above like it was around 1cm. It would seem that the threshold for change must be at least the instrument error, and that this should be applied for both accumulation and erosion. It is not reasonably possible to 'see' either erosion or deposition of less than the instrument

error.

Page 6. Line 3-5 " Due to the absence of reliable methods... ERA-I is one of the most reliable sources of information" What? There's no meaningful observations and therefore a model is reliable? This is pretty suspect logic. The winds don't even agree that well, in magnitude, and they are considerably easier to model than precip. Please revise this statement.

Page 6 line 20 (ac)cumulated Line 25 Condensation→ deposition Line 31 artefacts → artifacts Line 31 – please clarify what the contribution of instrument error, artifacts, and blowing snow returns is in this.

Page 7 Line 1 What role did the instrumentation presence play in increasing roughness?

Line 17 Shouldn't hiatus be anytime there not positive accumulation – erosion shouldn't actually be required to have a "hiatus"

Page 7 Line 26-27 This is too strong a conclusion based on this non-statistically significant, somewhat vague agreement. Change to ..."that despite the small extent of the RLS scanned area, the distribution of net annual accumulation (may be) representative of a wide(r) area."

Page 8 line 6 mathematical(ly)

Page 9 line 10 does not exists → (exist) Please confront: To what degree does the snow mix? If it should mix over some time period, perhaps accumulation is still somewhat by a continuous layer, rather than a "patchy" pattern. The layers just don't have the vertical resolution expected.

Page 9 line 22 – While this may be true for the ideal case, the reviewer does not believe that the scanner achieves a practical precision of 0.5 cm in surface positioning of real snow surfaces, particularly considering the interpolation involved in producing the gridded product.

Page 9 line 30 – Given the errors calling the 73 cm number "total deposition" is not a good terminology. It isn't really known what the total deposition is. Can you come up with a descriptor that doesn't misrepresent?

Line 32 rising→raising

Line 33 preventing description of the precise. . .

Line 33 delete "certainly" and replace with "likely" unless you have concrete evidence showing thescanner window was covered with snow.

Page 10 line 1 – speculative. Perhaps you can look up the wind speed and at least ensure it was well above the saltation threshold.

Page 10 line 2 rising → raising (Please get a native speaker to read. Not that this is poor quality writing, but there are a number of issues like this I've stopped correcting. There are also many unusual phrasings that are not exactly wrong, but hard to read).

Page 10 line 3-4 is this a dune or a drift caused by the laser station?

Figure 6. Sure looks like the tower tilted. What makes you sure it didn't? Is there independent confirmation of a large snow event?

Page 10 Line 12-15. Reviewer disagrees that this age of final deposition discontinuity necessarily results in large differences in snow chemistry. Some more thought should be given to the role of repeated redistribution in homogenization of the snowpack.

Page 13 Line 1-5 This sintering "increases over time at a rate increasing with temperature" – this doesn't seem to make sense, and it is not clear that sintering is faster (or more importantly creates more durable bonds more quickly) at higher temperatures. In contrast snow tends to sinter quite well under cold conditions with wind. The remaining two sentences after this are so speculative as to add little to the discussion.

Page 13 (4mm is the maximum of snow over the three year period in ERA-I). – Sure but that is irrelevant if you saw a much bigger accumulation in your data. ERA-I is just

a model.
* * *

---

## Referee Comment (RC2) · Anonymous Referee #2 · 29 Mar 2019

The manuscript "Observation of the process of snow accumulation on the Antarctic Plateau by time lapse laserscanning" by Picard et al. describes a very interesting, unique data set of daily repeated laser scans from the surface elevation at Dome C, Antarctica. Their is a lot of original, interesting analysis presented, which basically shows how complicated the assessment of the Antarctic mass balance is. The authors find that the accumulation is typically very patchy, multiple erosion/deposition cycles seem present before mass is finally "consolidated" into the snowpack. The snow surface shows high variability in terms of age of the snow. The manuscript is well written, and suited for The Cryosphere. There are a few issues that the authors should resolve before publication.

[Figure]

1) Treatment of "noise" is inconsistent, in any case in its explanation:

- p4,l6-7: I think it should be discussed how this accuracy changed due to the increase in installation height. I'm not convinced that the result from the 2016 study can just be applied here for period 2 as well.

- p4,l12-14: ".... must be kept in mind for the analysis." This remark is not followed up in the rest of the manuscript. How exactly is it taken into account in the analysis?

- p8,l18: Note that p4, l6 mentions a vertical accuracy of < 1cm, and here it is suddenly claimed that it is 0.25 cm? Where does this number comes from?

- If the accuracy is 0.25 cm, I'm not convinced that the class 0 - 0.5 cm and -0.5 to 0 cm in Fig. 5 has any significant meaning. Similar to the volume calculation in p9,l30, the 0 cm^3 threshold seems to ignore any potential accuracy issues.

- An important point for me is that the reader should be given some sense of how the choice of thresholds influence the provided statistics. For example, the statistic in p.10,l14: "About 55% of the surface is younger than 100 days" How would this statistic change if a threshold of 0.25 cm would be chosen, instead of 1 cm (p5,l25). It's important that those kind of statistics are accompanied by some kind of error estimate. Maybe repeat the analysis with a threshold of 0.5 and 1.5, and express the range in brackes after each statistic.

- Apparently, surface sublimation is not detectable? Maybe this could be briefly discussed by the authors?

2) It has not been discussed how the tilt angle is accounted for:

- p7,l2-6: I'm not convinced that it makes sense to analyze the raw standard deviation. Has there been any correction for the installation angle? Here, and throughout the rest of the manuscript, I think it would be much better to subtract the overall slope first, before analyzing the standard deviation. It could well be that the slope is due to a tilt angle, rather than some large scale feature.

[Figure]

- p14,L22: "standard deviation of up to 8 cm" is a misleading statement, as there is a background slope. Again, I would suggest analyzing the data after removing the slope, as a tilt in the mast with the laser scanner can produce a bias in standard deviation. For example, the increase in standard deviation from 4 cm in the beginning of the observation period to 8 cm towards the end could as well be explained by tilting of the mast with the laser scanner.

Other remarks:

p9,l1-2: Please explain why these dates were chosen.

p6,l1: As far as I know the literature, ERA-I is known to underestimate the SMB in the interior of Antarctica, see for example the Wang et al. 2016 paper. The authors even conclude that this is the case (p12,l20-22). So even though the overall SMB of Antarctica is correct, including high SMB coastal areas, we can expect ERA-I to underestimate the SMB at Dome C. That's a crucial point of information here, and deserves some discussion at this point.

p6,l15 and p8,l30: It should be better substantiated that this is not a measurement error. It's a very strange event. p8,l30: is ERA-I giving any strange weather patterns during this period? It's not only the accumulation that is strange, also the strong decrease in surface elevation afterwards, first steep decrease, then flattening out towards the end of the installation period 1, is something very out of the ordinary given the data presented for period 2a and 2b. Could the authors report in more detail what happened here, when looking at the individual scans?

Minor comments:

p3,l25: specify typical "unfavourable" conditions.

p3,l32: What are "reduced motors"?

p6,l20: The Petit, 1982 reference is a little bit out of place here, given that ERA-I is much more recent. How does the Petit, 1982 reference relates to ERA-I?

p12,l4: "patches on their way windward" I don't understand, should not be written that the patches migrate downwind?

p15,l10: superfluous "

Fig. 1: Please provide a detailed photo of the lasermeter. It's impossible to see now in the figure.

Fig. 2: Please provide a legend. The colours seem to slightly change between Period 1, 2a and 2b. Is this on purpose? If so, please, explain in the figure caption.

Fig. 6: Isn't the color bar legend "Daily accumulation" mistaken? It rather is the total accumulation over the 14 day period, I assume.

---

## Author Comment (AC1) · 11 May 2019

**General response**

We would like to thank both reviewers for the great attention they have shown in reading and commenting the paper. We understand their suspicion about the precision and stability of the laserscan dataset, and the potential impact on our conclusions. We have carefully addressed this point throughmany changes anda key addition that should hopefully solve many of the comments is an entire section dedicated to uncertainty evaluation. Such a section was already present in Picard et al. 2016, the first study about the RLS instrument, but we acknowledge that it is a valuable addition to this paper because of the longer time-series and the difference in the setup. And the section indeed shows that the second period of measurements has larger errors due to the higher setup position of the RLS.

We do agree that this dataset contains errors and artifacts that are close to the phenomena we try to detect and analyse. Nevertheless, we have described only the phenomena that are above the instrumental error limit and we have strengthened the statistical analysis or provided indications on the sensitivity in two sections: 1) for the computation of the cumulative positive/negative accumulation, we have devised a noise reduction processing step, and 2) for the age, we have used the algorithm with different thresholds and indicated the results.

However, it is worth noting that the error limit is difficult to estimate (because itself is affected by large uncertainties) and it is difficult to estimate the impact of this error in a rigorous statistical way for all the results. This is the reason why we focused on large patches (> 2cm), why the choices of the "thresholds" are quite constrained, and why sublimation is not addressed in this paper.

Despite these artifacts, we believe that this unique dataset is able to provide new qualitative and even semi-quantitative information on the accumulation process at small scale.

**Response to RC1**

The authors show a very interesting laser scanner dataset capturing the patchy accumulation of snow at Dome C. The analysis and discussion correctly highlight the issues this accumulation pattern may create for interpretation of presumed horizontal stratigraphy. The paper is of high quality and merits publication after a few issues are fixed.

1. Is the major accumulation event real or did the scanner tower tip. Convince your reader/reviewer of your conclusion.

2. Errors of the scanner precision under real world applications and errors added by interpolation process are underrepresented in the analysis and not used as benchmarks in deciding when erosion or accumulation is detected. They could be used to argue for statistical certainty of erosion or accumulation.

3. Some more thought should be given to the role of repeated redistribution in homogenization of the snowpack. The authors present the patchy accumulation as if it assures signals from short lived events (e.g. eruption ash) will be absent from many locations. It is also true that the redistribution vertically mixes short lived events, however, and may after many redistributions, place evidence of them at most locations.

We have addressed these three points as explained below in the detailed comments of the reviewer.

**Minor corrections follow:**

Page 2 Line 3 – "model's skills" check – did you mean model skill? models' skill

Corrected

Line 5 – condensation is transition to liquid from vapor. Deposition is phase transition to solid from vapor. Deposition is likely what is meant.

Corrected. French physicist use "condensation" for vapor to any condensed state (liquid or solid).

Line 7 – maybe clarify here that sublimation of blowing snow possible too, not just "from the surface"?

Blowing snow is addressed at the end of this paragraph. We have slightly modified this whole paragraph based on comments from Charles Amory. This should also make clearer that sublimation during blowing snow is addressed.

Line 14 – . . . scale of interest get(s). . .

Corrected

Line 16-18 – It is not clear whether erosion or deposition is the predominant cause of roughness, or which increases the roughness more. Over time, the authors note that the roughness remains nearly constant. Perhaps remove the statement that erosion is "generally increasing the roughness"

Done

Page 3 Line 7 The approach was quite successful (in reproducing) the. . .
Done

Line 15 delete space before ?

Done

Line 29 consider clarifying . . .. "in high (natural) light illumination"

We have changed to "We also found that despite a design for outdoor operations, the lasermeter performance **during the daytime** was greatly improved by adding a band-pass optical filter"

Line 32 ". . . enabling operations (at temperatures as low as -80C). . ."

Done

Page 4 Line 9-11 The criterial of 6 cm here, combined with an interpolation to 2 and 3 cm grids, respectively could be generating a considerable amount of 'made up' data. Can you defend this better for your reader? How many points did not have an actual observation within 1 or 1.5cm radius (the interpolated point spacing)? What is the total number of points before and after interpolation, particularly in low density areas of the survey?

We have checked more carefully the impact of the interpolation. We have first tried to reduced the 6 cm distance, but this increases the number of gaps too much, which reduces the visual quality of maps (because the eye is very sensitive to gaps, more than statistics). We have then, as suggested by the reviewer, computed the distance between each grid point and the nearest measurements (over a subset of 23 scans in the beginning of the period 2, totaling 3 million grid points). We have found that 93% of the points have at least one measurement within the grid spacing (3 cm, during the period 2) and 62% have at least one measurement within half a grid spacing (1.5 cm). We conclude that the effective resolution is very close to 3 cm horizontally and that the statistical impact of the 6 cm criteria is weak.

The text has been modified accordingly:
"The grid spacing is set to 2 and 3 cm respectively for the periods 1 and 2, in relationship to the different setup heights. To avoid filling large gaps with the bilinear interpolation, a grid point was attributed a valid z value only if at least one measurement was taken within twice the grid spacing (i.e. 4 and 6 cm for the periods 1 and 2 respectively) around it. In practice, we found that 93% (resp. 62%) of the grid points have at least one measurement within one (resp. half) grid spacing."

Two related remarks:
- we have adapted the distance of 6 cm to the resolution, now it is twice the resolution and rerun all the gridding.
- when checking this issue, we noticed that the first filter not only selects points based on their elevation z as written in the paper,  but also removes any point with less than 2 neighbors within a distance of 5 cm. We have now added this information in the paper.

Page 5 line 20-30 It is unclear in this discussion if the threshold for positive change is also applied to negative change. The discussion also does not incorporate any handling of instrument surface position detection error, which sounded above like it was around 1cm. It would seem that the threshold for change must be at least the instrument error, and that this should be applied for both accumulation and erosion. It is not reasonably possible to 'see' either erosion or deposition of less than the instrument error.

We have improved the method and results sections following reviewer comments:
- We have made clearer and explicit the thresholds for both erosion and accumulation. We have added mathematical formulation for the three cases (accumulation, erosion, no change).
- We have run the algorithm not only with the original settings (threshold 1 cm for accumulation and 0 cm for erosion) but also with symmetrical thresholds (0.5 cm and -0.5 cm). The differences are presented in the section "Age of the snow on the surface". The numbers are different, but nevertheless they give completely original information on the accumulation process and a concrete representation. These differences do not affect our conclusions.

We have not varied the thresholds over a wide range for the reasons already discussed above, there is a tradeoff between a threshold high enough above noise level, and small enough compared to the typical daily accumulation to make the detection of change effective. And the tradeoff is difficult because the instrument could  hardly be more precise, and on the other hand, Dome C accumulation is really weak.

In summary, we have largely changed the Sections 2.2, 2.3 and 3.5 to address this comment.

Page 6. Line 3-5 " Due to the absence of reliable methods. . . ERA-I is one of the
most reliable sources of information" What? There's no meaningful observations and
therefore a model is reliable? This is pretty suspect logic. The winds don't even agree
that well, in magnitude, and they are considerably easier to model than precip. Please
revise this statement.

We mean that when no observation is available in a place, relying on physical laws validated elsewhere with many observations is the best option. We agree with the reviewer that there is huge uncertainty associated to precipitation forecast in Antarctica. The result section and Fig 3 makes clear that ERA-I precipitation are far from our observations and the absence of significant correlation with surface change is not misinterpreted.

Our statement  "one of the most reliable sources of information to date (+2 references)" is already moderated compared to a typical statement expected in this kind of sentence (e.g. "the most accurate precipitation amount") and we have added "though it is known to underestimate accumulation near Dome C (Genthon et al. 2015, Wang et al. 2016)", which is repeated in the result section.

Page 6 line 20 (ac)cumulated

We have changed to "cumulative".

Line 25 Condensation→ deposition

Done

Line 31 artefacts → artifacts

Done

Line 31 – please clarify what the contribution of instrument error, artifacts, and blowing snow returns is in this.

We have added the information on how these numbers compare to the uncertainties now presented in the new Section dedicated to uncertainties.

"In particular it remained constant during the shift period 1 and 2a, despite a three-fold increase of the scanned area. These numbers are significant compared the instrumental error (Section 2.2), except during the last period (2b), we believe the standard deviation is unreliable, probably affected by the failures of the RLS."

Page 7 Line 1 What role did the instrumentation presence play in increasing roughness?

It is difficult to answer. We have tried to minimize the impact of the instrument, the structure is very thin (as seen in Figure 1) and it is downwind for the prevailing winds (coming from the South/ South-East, the South arrow is shown in Figure 8).

Line 17 Shouldn't hiatus be anytime there not positive accumulation – erosion shouldn't actually be required to have a "hiatus"

We do not understand the comment. We have removed the term hiatus from this page.

Page 7 Line 26-27 This is too strong a conclusion based on this non-statistically significant, somewhat vague agreement. Change to . . ."that despite the small extent of the RLS scanned area, the distribution of net annual accumulation (may be) representative
of a wide(r) area."

Done

Page 8 line 6 mathematical(ly)

Done

Page 9 line 10 does not exists → (exist) Please confront: To what degree does the snow mix? If it should mix over some time period, perhaps accumulation is still somewhat by a continuous layer, rather than a "patchy" pattern. The layers just don't have the vertical resolution expected.

We have add snow mixing in the discussion as suggested by the reviewer (see further comment) but we indeed can not address this question with the RLS data. However, we are not sure to understand how this relates / impacts the text around line 10.

Page 9 line 22 – While this may be true for the ideal case, the reviewer does not believe that the scanner achieves a practical precision of 0.5 cm in surface positioning of real snow surfaces, particularly considering the interpolation involved in producing the gridded product.

We have provided evidence in the new Section 2.2 dedicated on the uncertainties to show that the practical precision (not the accuracy) is below 0.5 cm on average. We have also improved the question of the interpolation in Section 2.3 with quantitative information.

Page 9 line 30 – Given the errors calling the 73 cm number "total deposition" is not a good terminology. It isn't really known what the total deposition is. Can you come up with a descriptor that doesn't misrepresent?

We have removed the term "total deposition" from the transition sentence at the end of the Section. The sentence now reads: "To further explore the accumulation process, in the next section we focus on the "major events" which amount to 22cm". We have also improved the calculation of the cumulative accumulation by reducing the noise with a statistical method. This leads to decreasing the value of 73 cm to 55 cm that we believe is now more robust. Even though this is still subject to caution, we believe the cumulative accumulation analysis in this section provides interesting information on the accumulation process.

Line 32 rising→raising

Done

Line 33 preventing description of the precise. . .

Done

Line 33 delete "certainly" and replace with "likely" unless you have concrete evidence showing the scanner window was covered with snow.

Done

Page 10 line 1 – speculative. Perhaps you can look up the wind speed and at least ensure it was well above the saltation threshold.

It is not clear which line it refer to. Based on the "rising" error found at page 10 and line 2, we deduced this comment is about "sintering". We removed this statement (as also suggested in a next comment).

Page 10 line 2 rising → raising (Please get a native speaker to read. Not that this is poor quality writing, but there are a number of issues like this I've stopped correcting. There are also many unusual phrasings that are not exactly wrong, but hard to read).

The English native co-author has read again the paper.

Page 10 line 3-4 is this a dune or a drift caused by the laser station?
Figure 6. Sure looks like the tower tilted. What makes you sure it didn't? Is there independent confirmation of a large snow event?

We have not detailed this event here because it was described in Picard et al. 2016. We also discussed the stability of the structure in that paper. We have now added new information on the stability of the structure (Fig 2 and Section 2.2) demonstrating that the change is significant compared to the stability deduced from the sphere. It is therefore a dune. Page 10, line 3-4 we have added: "We are confident that this is not an artifact because the surface elevation remained affected by this event for months, and the spheres confirm the stability of the setup."

Page 10 Line 12-15. Reviewer disagrees that this age of final deposition discontinuity necessarily results in large differences in snow chemistry. Some more thought should be given to the role of repeated redistribution in homogenization of the snowpack.

We have indeed overlooked this aspect and the complexity of how the deposition combines with our findings. We propose to complement the paragraph about ice cores with the Review's idea. From the sentence "They found that volcanic events were missing in 30% of the cores on average and that the flux uncertainty reached 65% when a single core was used." we have added:  " Our dataset is however insufficient to make a precise comparison with these statistics because the presence of a tracer in a snow patch depends on the date of first precipitation of that patch, which is unknown, and we can only estimate the date of its last deposition. Moreover considering that erosion occurs for 15 days on average (and up to 129 days) after deposition and considering that snow can be remobilized several times, a patch is in fact composed of snow precipitated over a time window of days to potentially years before settling. This tends to homogenize the presence of a tracer."

We have also amended the following sentence : "With the patchy accumulation and with snow age on the surface spanning at least one year, it is clear that some precipitation events, volcanic eruptions, or nuclear events may not be recorded everywhere, at least not with the same intensity and depending on the duration of the events, and the mode of deposition (dry or wet). ".

Page 13 Line 1-5 This sintering "increases over time at a rate increasing with temperature" – this doesn't seem to make sense, and it is not clear that sintering is faster (or more importantly creates more durable bonds more quickly) at higher temperatures. In contrast snow tends to sinter quite well under cold conditions with wind. The remaining two sentences after this are so speculative as to add little to the discussion.

We have removed the sentences as suggested.

Page 13 (4mm is the maximum of snow over the three year period in ERA-I). – Sure but that is irrelevant if you saw a much bigger accumulation in your data. ERA-I is just a model.

The statement here relies on the difference of order of magnitude (cm versus mm) and not on precise values. We have added "e.g." to underline that our conclusion does not rely on the value in parenthesis. "(e.g. 4 mm is the maximum of snow over the three year period in ERA-I)".

**Response to RC2**

The manuscript "Observation of the process of snow accumulation on the Antarctic Plateau by time lapse laserscanning" by Picard et al. describes a very interesting, unique data set of daily repeated laser scans from the surface elevation at Dome C, Antarctica. Their is a lot of original, interesting analysis presented, which basically shows how complicated the assessment of the Antarctic mass balance is. The authors find that the accumulation is typically very patchy, multiple erosion/deposition cycles seem present before mass is finally "consolidated" into the snowpack. The snow surface shows high variability in terms of age of the snow. The manuscript is well written, and suited for The Cryosphere. There are a few issues that the authors should resolve before publication.

**1) Treatment of "noise" is inconsistent, in any case in its explanation:**
- p4,l6-7: I think it should be discussed how this accuracy changed due to the increase in installation height. I'm not convinced that the result from the 2016 study can just be applied here for period 2 as well.

This is now addressed in the new section 2.2. The precision is indeed affected by the higher setup.

- p4,l12-14: ".... must be kept in mind for the analysis." This remark is not followed up in the rest of the manuscript. How exactly is it taken into account in the analysis?

Our analysis has been greatly influenced by this limitation in our dataset and the manuscript only shows what we consider to be the most robust / interesting results. For instance, we have taken into account this problem as follows:
- in the way to compute the age or the snowpack structure, the algorithms are relatively robust to the missing data. We have added new results using the age algorithm with different thresholds.
- in the focus given to period 2a which has the highest rate of true daily increments (90%) and the use of the other periods only when the increments has a lesser importance. We have added a noise reduction in this computation.
- in a few remarks such as "(called daily accumulation here despite the irregular sampling in time) "

We have not, however, tried to systematically estimate the bias induced by this limitation (i.e. with proper statistical modeling), because this is too difficult and probably too inaccurate compared to the benefit for the results and the main conclusions of the paper.

- p8,l18: Note that p4, l6 mentions a vertical accuracy of < 1cm, and here it is suddenly claimed that it is 0.25 cm? Where does this number comes from?

We provide three kind of values:
- the absolute accuracy / long term stability which is indeed of the order of 1 cm (for period 1). This was largely addressed in the first about snow depth (Picard et al. 2016), but it is not relevant for the present study which focuses on elevation changes.
- the precision in the scan area which is about 0.5 cm.
- the precision of the difference between successive measurements, which is of the order of 0.2-0.4 cm depending on the period. These later numbers are the most relevant for the daily accumulation, the reason why p8, l18 gives a value of 0.25cm (now 0.2-0.4cm).
The new Section 2.2 hopefully better explains these differences.

- If the accuracy is 0.25 cm, I'm not convinced that the class 0 - 0.5 cm and -0.5 to 0 cm in Fig. 5 has any significant meaning. Similar to the volume calculation in p9,l30, the 0 cmˆ3 threshold seems to ignore any potential accuracy issues.

We have changed how the data are processed and applied a noise reduction to limit the problem described by the reviewer. This changes the Fig 5 (now Fig 6) quite significantly but does not affect the overall message that snow is remobilized several times before settling. The volume calculation for the patches is over a very large number of measurement points, it  can not be affected by precision.

- An important point for me is that the reader should be given some sense of how the choice of thresholds influence the provided statistics. For example, the statistic in p.10,l14: "About 55% of the surface is younger than 100 days" How would this statistic change if a threshold of 0.25 cm would be chosen, instead of 1 cm (p5, l25). It's important that those kind of statistics are accompanied by some kind of error estimate. Maybe repeat the analysis with a threshold of 0.5 and 1.5, and express the range in brackes after each statistic.

We have added the information suggested by the reviewer using different thresholds and shown the result in Figure 11 (now Fig 12). It is worth highlighting however that the choice of the threshold is constrained, with on one hand the precision being 0.4 cm and on the other hand, the typical daily accumulation being much less than 2 cm at most. Using 1.5cm as suggested would certainly change the results, because it would change the interpretation of the algorithm and results, and would not be related to "a kind of error estimate".

Most importantly by changing Fig 12 and all the additions done around the precision throughout the paper, we believe we have raised the awareness about this question and hopefully avoided over-interpretation by the readers of the quantitative results.

- Apparently, surface sublimation is not detectable? Maybe this could be briefly discussed by the authors?

We agree that sublimation is probably not detectable, not only because of the precision, but also because it does not necessary result in a change of the surface elevation but rather causes a density decrease of the superficial snow. Unless the reviewer sees a particular point in the text where to add this information, we prefer to avoid non essential additions, and keep the focus on the marked accumulation processes that are above the instrument error.

**2) It has not been discussed how the tilt angle is accounted for:**
- p7,l2-6: I'm not convinced that it makes sense to analyze the raw standard deviation.
Has there been any correction for the installation angle? Here, and throughout the rest of the manuscript, I think it would be much better to subtract the overall slope first, before analyzing the standard deviation. It could well be that the slope is due to a tilt angle, rather than some large scale feature.

We have added the information to show that the major local slope is due to a dune, not to a tilt of the mast in Section 2.2. The leveling of the laserscan has been monitored as much as possible with the spheres. The conclusion is that both standard deviation values, with and without the main slope, are reliable. In addition, we believe they both are informative: The total (raw) standard deviation (ie. including the slope) is to be compared with annual accumulation and for aerodynamic and geometrical roughness because it includes all the spatial  scales, at least all the scales accessible with the RLS. We also presented the local standard deviation (ie. slope removed) to provide a different information, related to shorter scales roughness.

It seems that the reviewer's remark is mostly driven by the reliability (which we have now better addressed) and maybe also by the general idea that the standard deviation is more intuitive for a stationary signal. Therefore, to avoid such confusion, we have renamed "standard deviation of surface elevation" to "RMS height", which is more common in the remote sensing community.

We would at last like to emphasis that the purpose of this calculation is to draw an important conclusion of the paper: the terrain undulation characteristic height (here calculated as the RMS height / standard deviation) is significant compared to the annual accumulation, which is a typical trait of the Antarctic Plateau, as opposed to the Alpine regions for instance. This conclusion does not depend on the exact metrics used to estimate this terrain undulation characteristic height.

- p14,L22: "standard deviation of up to 8 cm" is a misleading statement, as there is a background slope. Again, I would suggest analyzing the data after removing the slope, as a tilt in the mast with the laser scanner can produce a bias in standard deviation. For example, the increase in standard deviation from 4 cm in the beginning of the observation period to 8 cm towards the end could as well be explained by tilting of the mast with the laser scanner.

This is now addressed (refer to our response to the previous comment).

**Other remarks:**
p9,l1-2: Please explain why these dates were chosen.

We have changed the sentence to make clear that these dates are not special, they have been chosen in contrast to the accumulation map of the major event (Figure 6) and to illustrate the patchy accumulation. The beginning of the paragraph is now: "In contrast to this rare event, accumulation maps usually look different. This is illustrated by the accumulation maps between 28 and 29 May 2016 and between 29 and 30 May 2016 (Figure 7)."

p6,l1: As far as I know the literature, ERA-I is known to underestimate the SMB in the interior of Antarctica, see for example the Wang et al. 2016 paper. The authors even conclude that this is the case (p12,l20-22). So even though the overall SMB of Antarctica is correct, including high SMB coastal areas, we can expect ERA-I to underestimate the SMB at Dome C. That's a crucial point of information here, and deserves some discussion at this point.

We have added "though it is known to underestimate accumulation near Dome C (Genthon et al. 2015, Wang et al. 2016)"

p6,l15 and p8,l30: It should be better substantiated that this is not a measurement error. It's a very strange event.

We have added the information to show that it is reliable. This event is probably local, and probably not so strange as shown on the photographs (and see next comment).

p8,l30: is ERA-I giving any strange weather patterns during this period? It's not only the accumulation that is strange, also the strong decrease in surface elevation afterwards, first steep decrease, then flattening out towards the end of the installation period 1, is something very out of the ordinary given the data presented for period 2a and 2b. Could the authors report in more detail what happened here, when looking at the individual scans?
ERA-I predicts a snowfall, but nothing exceptional in terms of magnitude as shown in Fig 2 (the green line). As explained before, this event was investigated in Picard et al. 2016 and we have chosen not to analyse it here because its scale is larger than what is accessible with the RLS. We also do not have the full sequence of events as explained in the section. Our plan is to set the RLS higher in the coming years to cover this kind of object.

The decrease of the curve shows the erosion occurring after the event. This is quite common as shown in the section about the patches, except that here it concerns a large proportion of the scanned area, and therefore is

visible even on the overall mean surface elevation (Fig 4), whilst the patches are usually much smaller and often compensated by nearby erosion.

Even though we cannot prove it, we believe that it is a decameter-scale dune that is about half in / half out of the scanned area. It would be similar to those evoked in Picard et al. 2014 or described precisely in Sommer et al. 2018) and similar to the barchan shown in Figure 13 (now Fig 14). This would not be due to a particularly large snowfall. We have slightly changed the text by "It is also worth mentioning that the patches studied throughout this paper are smaller in general than the accumulation pattern that appeared in 4-16 July 2015 (Fig. 7). The latter is probably more similar to the barchan dunes visible in the photographies in Figure 14 and evoked in Picard et al. 2014 and Sommer et al. 2018. The former are also well aligned with the wind direction (longitudinal bedform) while the latter have tails elongated at ~45° with respect to the wind direction (Filhol et al. 2015). These clearly are different objects, with different size (meter versus decameter) and different dynamics (daily versus yearly)."

**Minor comments:**
p3,l25: specify typical "unfavourable" conditions.

We have added: " (high luminosity, blowing snow, rapidly changing surface)"

p3,l32: What are "reduced motors"?

Sorry a too direct translation from French. We have removed this unimportant information.

p6,l20: The Petit, 1982 reference is a little bit out of place here, given that ERA-I is much more recent. How does the Petit, 1982 reference relates to ERA-I?

Petit gives Dome C accumulation reference, but we have removed it here as the sentence is about snowfall not accumulation and the following sentence better addresses the problem of underestimation

p12,l4: "patches on their way windward" I don't understand, should not be written that the patches migrate downwind?

Exact. Corrected.

p15,l10: superfluous "

done

Fig. 1: Please provide a detailed photo of the lasermeter. It's impossible to see now in the figure.

To avoid duplication, we have added the reference to Picard et al. 2016, which is the main paper describing the instrument.

Fig. 2: Please provide a legend. The colours seem to slightly change between Period 1, 2a and 2b. Is this on purpose? If so, please, explain in the figure caption.

The purpose was intially to distinguish the 3 periods but they have subsequently been marked on the graph. We've removed these small differences.

Fig. 6: Isn't the color bar legend "Daily accumulation" mistaken? It rather is the total accumulation over the 14 day period, I assume.

Yes. Corrected. We have also found that the data for 16 July 2015, discarded in the first version of the paper, were in fact correct. The map is now from 4 to 16 July, but it looks very similar to that from 4 to 17 July because the main accumulation occurred before the end of this period. The text is updated as well.